# Prioritizing Parkinson's disease genes using population-scale transcriptomic data

Yang I. Li[1], Garrett Wong[2], Jack Humphrey [3,4] & Towfique Raj[2]

Genome-wide association studies (GWAS) have identified over 41 susceptibility loci associated with Parkinson's Disease (PD) but identifying putative causal genes and the underlying mechanisms remains challenging. Here, we leverage large-scale transcriptomic datasets to prioritize genes that are likely to affect PD by using a transcriptome-wide association study (TWAS) approach. Using this approach, we identify 66 gene associations whose predicted expression or splicing levels in dorsolateral prefrontal cortex (DLFPC) and peripheral monocytes are significantly associated with PD risk. We uncover many novel genes associated with PD but also novel mechanisms for known associations such as *MAPT*, for which we find that variation in exon 3 splicing explains the common genetic association. Genes identified in our analyses belong to the same or related pathways including lysosomal and innate immune function. Overall, our study provides a strong foundation for further mechanistic studies that will elucidate the molecular drivers of PD.

[1] Section of Genetic Medicine, Department of Medicine, and Department of Human Genetics, University of Chicago, Chicago 60637 IL, USA. [2] Departments of Neuroscience, and Genetics and Genomic Sciences, Ronald M. Loeb Center for Alzheimer's disease, Icahn School of Medicine at Mount Sinai, New York 10029 NY, USA. [3] UCL Genetics Institute, Gower Street, London WC1E 6BT, UK. [4] Department of Neurodegenerative Disease, UCL Institute of Neurology, London WC1E 6BT, UK. These authors contributed equally: Yang I. Li, Garrett Wong. Correspondence and requests for materials should be addressed to T.R. (email: towfique.raj@mssm.edu)

Parkinson's disease (PD) is the second most common neurodegenerative disorder after Alzheimer's disease (AD). PD is characterized by the formation of intracytoplasmic inclusions known as Lewy bodies containing α-synuclein (α-syn) and by the loss of dopaminergic neurons primarily in the substantia nigra[1,2]. Over the past decade, a number of genetic susceptibility factors have been identified for PD, including nine genes linked to heritable, monogenic forms of PD[2]. More recently, over 41 genetic susceptibility loci have been associated with late-onset PD in the largest genome-wide association studies (GWAS) meta-analysis of PD to date[3]. Within these risk loci, a few genes have been identified as potentially causal, but for the majority of loci, it is not yet known which genes underlie PD risk. More generally, it is currently unclear whether PD risk genes identified thus far by GWAS studies belong to coherent pathways such as those involved in lysosomal and autophagy function as was previously suggested[3].

The integration of large-scale functional genomics data with GWAS has been a powerful approach to characterize the functional effects of associated variants[3,4]. PD is generally considered to be a disease involving neurons[1,2]. As such, many studies have attempted to use functional genomics data in post-mortem human brains to ascribe function to PD risk variants[5–7]. However, previous studies were underpowered due to limited sample sizes and the high cellular heterogeneity of brain tissues, but in few cases, they have led to novel biological hypotheses about the mechanisms underlying PD-associated genetic associations. For instance, at the *MAPT* locus, a well-characterized inversion at 17q21 marks the PD-associated H2 haplotype, which has been reported to be associated with exon 10 exclusion[8], *MAPT* exon 3 inclusion[9], and also increased total *MAPT* expression[8]. While these reports suggest that *MAPT* is involved in PD, it is unknown what specific mechanism drives the association to PD and whether the H1/H2 haplotypes are also associated with differential expression and/or splicing of other genes in the locus.

Beyond neuronal cells, immune cells have also been suggested to play a role in PD[10]. For example, we have previously used genomic approaches to show that PD-associated loci are enriched in expression quantitative loci (eQTLs) from peripheral monocytes but not CD4+ T cells[10]. Similarly, recent studies used cell-type-specific functional annotations to show that genes in PD GWAS loci are preferentially expressed in CD14+ monocytes[11,12]. These findings support the involvement of the innate-immune cells, including peripheral monocytes and central nervous system (CNS) microglia, in the etiology of PD.

Here, we use a transcriptome-wide association study (TWAS) approach to prioritize candidate PD genes and to better understand the primary mechanisms that underlie PD genetic risk factors. Previously, TWAS used expression quantitative trait loci (eQTL) data to impute RNA expression levels onto large cohorts of individuals from GWASs to identify putative genes involved in complex autoimmune diseases[13] or cardio-metabolic traits[14]. We have recently shown that genetic effects on RNA splicing, or splicing QTL (sQTLs), are likely primary mediators of genetic effects on complex disease at many GWAS loci[15]. Importantly, the genetic effects on RNA splicing are largely independent of that on RNA expression[15], and therefore we reason that they could help us identify additional links between disease-associated variants and candidate disease genes.

In this study, we consider both the genetic effects on RNA expression and splicing to prioritize disease-relevant genes. To this end, we obtain PD GWAS data and large-scale transcriptomics data that are publicly available. We first use broad atlases of gene expression that include brain[16,17] and immune cell types[17,18] to find tissues or cell types with specifically expressed genes (SEGs) for PD susceptibility loci. We then leverage large-scale transcriptomic data from primary monocytes[10,19,20] and a large-scale dorsolateral prefrontal cortex (DLPFC) dataset[21] to perform a TWAS of PD. We prioritize 66 genes whose predicted expression or splicing levels in peripheral monocytes cells and in DLPFC are significantly associated with PD risk. Overall, our study suggests that a TWAS approach considering both genetic effects on RNA expression and splicing is a powerful method to identify specific genes and mechanisms at each GWAS locus as determinants of PD risk.

## Results

**Heritability enrichment of expressed genes identifies PD-relevant tissues.** To obtain a better understanding of how genetic variants affect PD risk, we first identified tissues or cell types likely to be relevant in PD pathology. To this end, we used linkage disequilibrium (LD) score regression for SEGs (LDSC-SEG), which is a computational method that identifies tissues in which genes with increased expression are enriched in single-nucleotide polymorphisms (SNPs) that tag an unexpectedly large amount of PD heritability[22]. When applied to 53 tissues from the genotype tissue expression (GTEx) project[16], we detected an enrichment at a 5% false discovery rate (FDR) threshold ($-\log_{10} p$ value >2.86) for six tissues including the amygdala, substantia nigra, anterior cingulate cortex, frontal cortex, hypothalamus, and cervical (C1) spinal cord (Fig. 1b). These findings can be replicated using a larger expression panel comprising of 152 cell types[17] (Supplementary Figure 1). In contrast, there is no enrichment of SNP heritability near genes expressed in CNS tissues for AD, a neurodegenerative disorder thought to share common etiologies with PD. The contrast in disease-associated variants enrichment in CNS tissues between PD and AD suggests that neuronal cell types are affected in fundamentally different ways in the two neurodegenerative diseases. Moreover, when we used LDSC-SEG on expression data from sorted primary mouse CNS cells[23], we found that neurons ($p = 0.024$, $t$-test) and oligodendrocytes ($p = 0.028$, $t$-test), but not astrocytes ($p = 0.91$, $t$-test), preferentially expressed genes enriched in SNP heritability for PD pathology (Supplementary Figure 2). Finally, we applied LDSC-SEG on an atlas of 291 mouse immunological cell types[18] in order to assess whether PD signal enrichment near genes specifically expressed in myeloid cells are higher than compared to other immune cell types, as previous results suggest[11,12] (Fig. 1b). However, again unlike for AD, where a marked enrichment for myeloid cell types was observed, the enrichment in PD is weak or absent, although it may be that specific cellular processes within myeloid cells are enriched for PD susceptibility genes[24].

We next asked whether PD-associated variants were enriched among molecular (including expression, splicing, histone marks, and DNA methylation) QTLs in DLPFC[27,28] and immune cell types[10,29] using GARFIELD[30] (Fig. 1c). GARFIELD uses functional annotations to assess whether GWAS signals are enriched for specific functions. Here we annotated each SNP by whether they were molecular QTLs in different datasets and cell types. To this end, we used eQTL and sQTL obtained from DLPFC RNA-sequencing (RNA-seq) samples from the Religious Orders Study (ROS) and Memory and Aging Project (MAP)[27] and Common-Mind Consortium (CMC)[21], which consists of 450 and 452 postmortem DLPFC, respectively. The eQTL and QTL for histone marks for immune cell types were obtained from the ImmVar Consortium ($n = 461$)[10], the BLUEPRINT Consortium ($n = 197$)[29], and a study by Fairfax et al.[19] ($n = 432$). As expected[15], we found that most molecular QTLs in both immune cell types and DLPFC were enriched in PD-associated variants, which implies a widespread gene regulatory impact for PD-associated variants. More interestingly, however, we found that genetic variants that

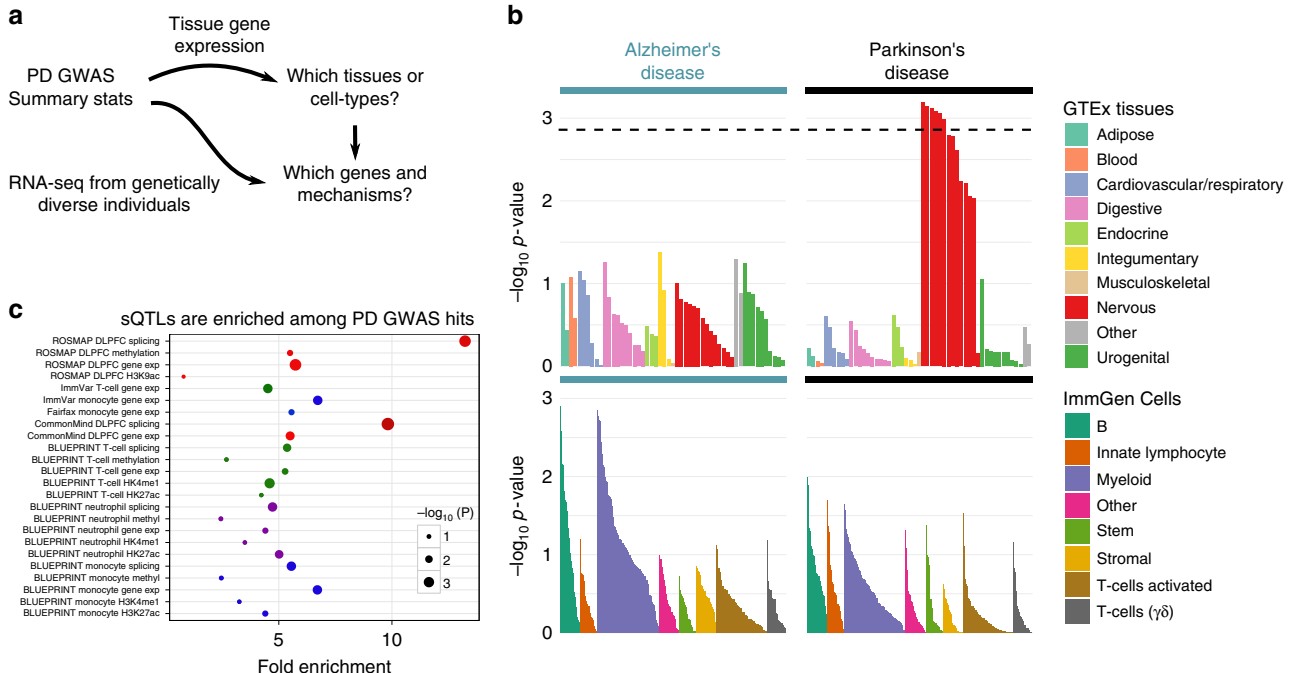

**Fig. 1** Enrichment of tissues and cell types in Parkinson's disease (PD). **a** Study design. Using PD genome-wide association studies (GWAS) summary statistics and gene expression datasets to: (1) identify tissues and cell types enriched in PD GWAS signal, and (2) to identify genes and mechanisms through which PD-associated loci act to affect disease risk. **b** Linkage disequilibrium (LD) score regression in specifically expressed genes (LD-SEG) analysis applied to late-onset AD (17,008 cases and 37,154 controls)[26] and PD (13,708 cases and 95,282 controls)[31] GWAS summary statistics. Top panel: Regions of the genome with specific expression in central nervous system (CNS) tissues are highly enriched for PD GWAS signal, among 53 tissues obtained from the Genotype-Tissue expression project (GTEx). Shown in red bars are CNS tissues. Six CNS tissues (amygdala, substantia nigra, anterior cingulate cortex, frontal cortex, hypothalamus, and cervical (C1) spinal cord) are significant at 5% false discovery rate (FDR) (dotted line). Bottom panel: LD-SEG analysis using immune cells from the ImmGen Consortium. **c** Enrichment of gene expression, splicing, methylation and chromatin quantitative loci (QTL) in PD GWAS (p value < 10−5) across tissues and cell types with GARFIELD. GARFIELD leverages GWAS signal with functional annotations to find features such as QTL annotations relevant to a phenotype of interest. All molecular QTLs were enriched in PD-associated variants, and that brain splicing QTLs (sQTLs) showed the strongest enrichment in PD-associated variants among all molecular QTLs

affect RNA splicing in DLPFC were highly enriched (14× fold enrichment) in variants associated with PD (at a p value cut-off <10−5; Fig. 1c), an observation that was shared in the two DLPFC datasets[21,27]. eQTLs in monocytes were also highly enriched among variants associated with PD (7× fold enrichment; Fig. 1c). Altogether, these results motivated us to consider both RNA splicing and RNA expression in DLPFC and monocytes as mediators of genetic effects on PD risk.

**TWAS of PD**. To identify and prioritize candidate PD genes, we performed TWAS[14] using summary-level data from a PD GWAS of 108,990 individuals of European ancestry (13,708 cases and 95,282 controls)[31] and transcriptome panels from peripheral monocytes and DLPFC. The TWAS approach uses information from the RNA expression or splicing measured in a reference panel and the PD GWAS summary statistics to evaluate the association between the genetic component of expression and PD status (see Methods). We built TWAS models using gene expression data from primary peripheral monocytes in three independent cohorts[10,19,20] and DLFPC RNA-seq data (n = 452) from CMC[21]. In addition to using gene expression to build predictive models, RNA-seq data from DLFPC allowed us to consider alternative splicing in PD by quantifying "percent spliced in" of splicing events[32]. However, we were unable to build a model with RNA splicing in monocytes as population-level RNA-seq data for this cell type was not available at the time of this study.

We used the FUSION software (see URLs) to estimate the heritability, build predictive models, and perform TWAS. For each reference panel, FUSION estimates the heritability of gene expression and alternative splicing explained by local SNPs (i.e., 1 Mb from TSS of each gene) using linear-mixed models[33]. Genes or splicing events that are nominally significant at $p < 0.01$ for SNP heritability ($cis\text{-}h_g^2$) are used for training predictive models (Supplementary Figure 3). FUSION fits four predictive linear models (see Methods) for every gene or intronic excision event using local SNPs as predictors. The models with the best cross-validation prediction accuracy are kept for prediction into GWAS (Supplementary Figures 4 and 5). In total, 17,798 tissue-specific models, including 4721 monocyte expression, 5383 DLPFC expression, and 7695 DLFPC alternatively spliced introns, were used for TWAS. The square of correlation ($R^2$) between predicted and observed gene expression levels normalized by corresponding $cis\text{-}h_g^2$ was calculated to measure prediction accuracy. Across all cohorts least absolute shrinkage and selection operator (LASSO) attained the best predictive performance, with 30% improvement in prediction $R^2$ over other models (Supplementary Figure 5). We found the average in-sample prediction accuracy ($R^2/cis\text{-}h_g^2$) to be 54, 72, and 73% for monocyte, DLFPC expression, and DLFPC splicing, respectively. These results are consistent previous TWAS analyses[14] and suggest that most of the signal in cis-regulated total expression and splicing levels is captured by the fitted models.

Using these TWAS models, we found that the expression of 29 genes is significantly associated with PD in monocytes (Fig. 2a),

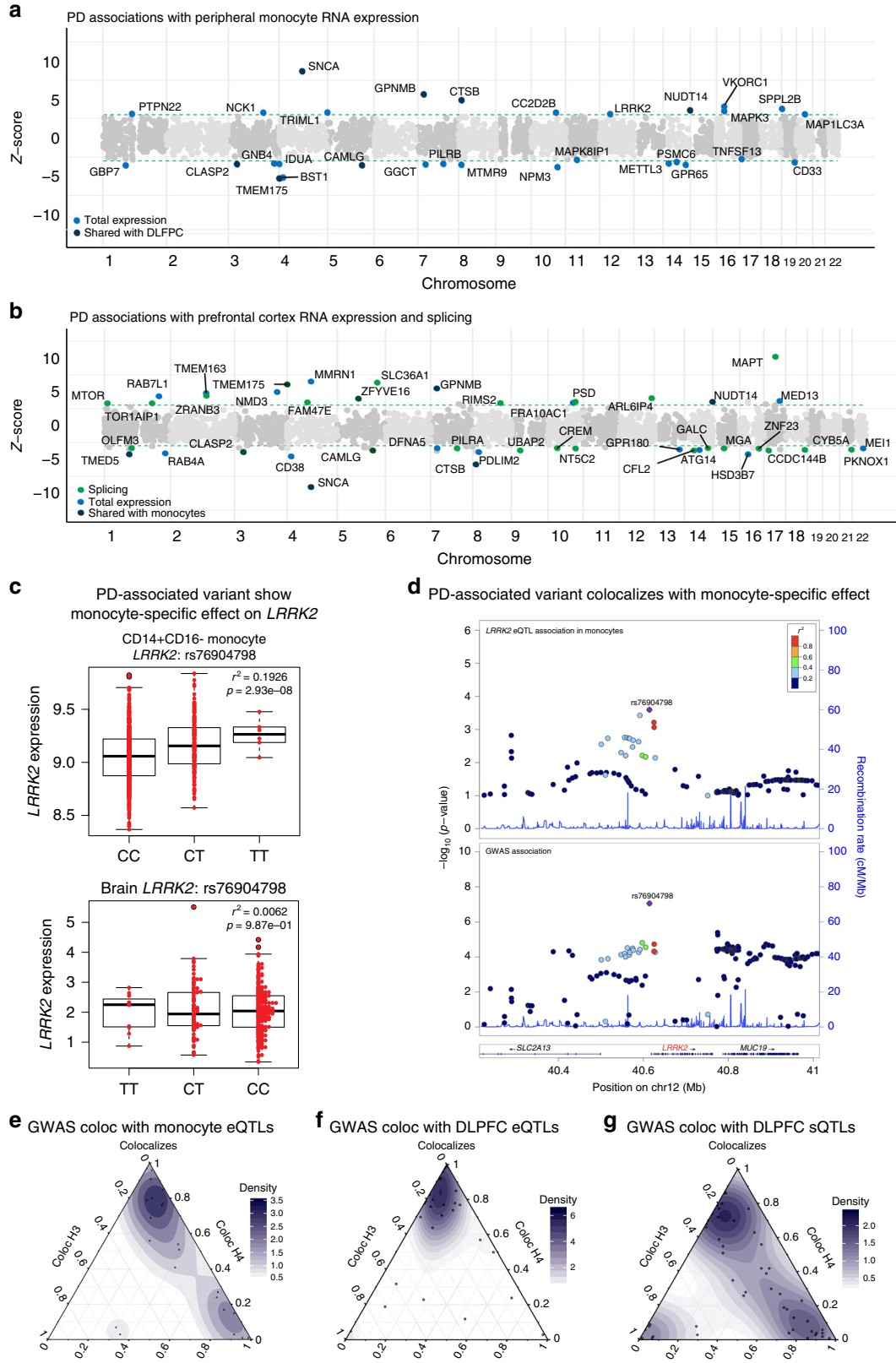

a PD associations with peripheral monocyte RNA expression

b PD associations with prefrontal cortex RNA expression and splicing

c PD-associated variant show monocyte-specific effect on *LRRK2*

d PD-associated variant colocalizes with monocyte-specific effect

e GWAS coloc with monocyte eQTLs

f GWAS coloc with DLPFC eQTLs

g GWAS coloc with DLPFC sQTLs

and that 44 genes are significantly associated with PD in DLPFC through RNA expression and/or splicing (FDR < .05) (Fig. 2b, Supplementary Data 1). Nineteen out of the 66 unique TWAS genes are in known PD GWAS susceptibility loci. Of the 29 genes associated with PD in monocytes, several genes were in novel PD

loci but previously associated with AD, for example, *CD33* and *PILRB*[26,34,35]. *CD33* protein level was previously shown to be affected by PD risk variants in monocytes[36], and *PILRB* is a binding partner for *TYROBP* (*DAP12*), the main regulator of the microglia network activated in AD[37]. The overlap between PD

**Fig. 2** Transcriptome-wide association study of Parkinson's disease (PD). **a** Manhattan plot of PD transcriptome-wide association study (TWAS) using gene expression models from peripheral monocytes[10,19,20]. Each point represents a single gene tested, with physical position plotted on the x-axis and Z-score of association between gene expression or intronic splicing with PD plotted on the y-axis. Colored (black or blue) points represent significant association to PD at 5% false discovery rate (FDR). Genes that are also identified with TWAS models from prefrontal cortex are labeled with black points. **b** Manhattan plot of PD TWAS using expression and splicing models from dorsolateral prefrontal cortex (CommonMind Consortium (CMC))[21]. Genes associated through variation in gene expression and splicing are labeled with blue and green points, respectively. Genes that are also identified with TWAS models from monocytes are labeled with black points. **c** Boxplot showing the association between a single-nucleotide polymorphism (SNP) (rs76904798) that tags the PD genome-wide association studies (GWAS) risk loci at *LRRK2* and gene expression level of *LRRK2* in monocytes (top) and dorsolateral prefrontal cortex (DLPFC) (bottom). rs76904798 is significantly associated with the expression level of *LRRK2* in monocytes, but not in DLFPC. **d** LocusZoom plot for the region surrounding *LRRK2* shows colocalization of the monocytes *LRRK2* expression quantitative loci (eQTL) (top) and PD GWAS association signal (bottom). **e–g** Ternary plots showing coloc posterior probabilities that TWAS loci found using RNA expression in monocytes, RNA expression in DLPFC, and RNA splicing in DLPFC, respectively, belong to different sharing configurations (colocalizing, independent, or underpowered). We considered H0 + H1 + H2 as evidence for the lack of test power. H0: no causal variant, H1: causal variant for PD GWAS only, H2: causal variant for QTL only, H3: two distinct causal variants, H4: one common causal variant

and AD risk genes supports the existence of shared genetic risk factors—which likely involve immune-mediated mechanisms—for the two neurodegenerative diseases[38,39].

In addition to new PD loci, we also observed significant associations with several genes previously known to play a role in PD, for example, *SNCA* and *LRRK2* (Supplementary Data 1). Interestingly, we found *SNCA* to be significantly associated with PD using both the monocyte and DLPFC TWAS models, but that *LRRK2* was only significantly associated with PD in monocytes. We, therefore, asked whether the *LRRK2* risk loci show evidence of cell-type-specific effects in monocytes, and not in cells from the CNS. Indeed, we found that the PD risk allele (rs76904798-T) at the *LRRK2* locus is associated with increased expression of *LRRK2* in monocytes (rs76904798, $p = 2.93 \times 10^{-8}$), but not in DLFPC ($p = 0.98$) (Fig. 2c). At this locus, the GWAS signal colocalizes with the eQTL signal in monocytes with posterior probability 0.99 (Fig. 2d; Supplementary Figure 6). Recent studies have shown that both *SNCA* and *LRRK2* are highly expressed in human microglia[39], and that the expression levels of these two genes are elevated in peripheral monocytes of PD patients compared to age-matched controls[40,41]. In addition, both *GPNMB* and *GBP7* were previously shown to be differentially expressed in PD brains vs. controls[42]. In total, seven genes (*SNCA*, *CLASP2*, *TMEM175*, *GPNMB*, *CTSB*, *CAMLG*, and *NUDT14*) were significant in both monocyte and DLPFC TWAS models. The finding that PD risk loci have cell-type-specific effects in monocytes also support the hypothesis that peripheral monocytes play a role in the progression of PD and/or may serve as a proxy for microglial activities within the brain.

To determine if the association signals for PD GWAS and expression (or splicing) are driven by the same causal variant, we next used coloc[43] to assess colocalization between the PD association signal at these TWAS loci and expression or splicing QTLs. We found that 10 of 29 loci, or 34%, had evidence of colocalization with monocyte eQTLs. In DLPFC, we found that 12 of 18 loci, or 67%, had evidence of colocalization with DLPFC eQTLs, and 4 of 26 loci, or 15%, had evidence of colocalization with DLPFC sQTLs (Supplementary Data 1). The significantly lower colocalization between GWAS signals and sQTLs compared to eQTLs is unexpected, as we found higher enrichment for PD GWAS signal among sQTLs than eQTLs (Figs. 1c, 2e–g). A plausible explanation is that our TWAS models for RNA splicing produced a higher false-positive rate than our TWAS model for RNA expression. However, another possibility is that genetic variants that affect RNA splicing tend to have smaller effects on complex traits or to be secondary associations. Approaches such as coloc depend on the alignment of strong effects both at the GWAS and QTL mapping levels, and thus they have limited ability to detect colocalization between weak GWAS and/or sQTL

signals. Indeed, when we plotted the posterior probabilities of our coloc analyses, we found that coloc performed well for eQTLs (Fig. 3e, f), assigning all probability density to H3 (independent signals) or H4 (colocalized signal). However, for a large number of sQTLs, coloc was underpowered to find evidence for colocalization (Fig. 2f), because it did not find evidence supporting the GWAS loci (H1), the sQTL association (H2), or neither (H0). Overall, our analysis indicates that the sQTL signal is independent of the GWAS signal for 5 loci (Fig. 2f) and colocalizes at 4 loci, but for the remaining 17 of the 26 loci that were associated with PD through RNA splicing, it is unknown whether the sQTL colocalizes with the PD GWAS signal or not.

We next sought to validate the results from our splicing TWAS analysis using a replication approach. We performed replication analyses of our TWAS in independent transcriptome[27] and GWAS datasets (a cohort of 23andMe research participants; 4124 cases and 62,037 controls[31]). We first assessed whether the DLPFC TWAS results could be replicated in an independent transcriptomic reference panel by performing TWAS using DLPFC RNA-seq data from ROSMAP ($n = 450$)[28]. Before validating the TWAS results in ROSMAP, we evaluated cross-cohort prediction of the genes and intronic splicing in CMC and ROSMAP. The average *cis*-heritability estimates between the ROSMAP and CMC were strongly correlated across genes and intronic splicing (Pearson's $r = 0.54$). The prediction accuracy between the two cohorts was also strongly correlated (Pearson's $r = 0.43$ and 0.16 for $R^2$ and $R^2/h_g^2$, respectively) (Supplementary Figures 7–8). The weaker correlation of the normalized prediction accuracy is likely due to due differences in the average *cis*-heritability estimates between ROSMAP (*cis*-$h_g^2 = 0.098$) and CMC (*cis*-$h_g^2 = 0.078$). To test the predictive consistency for models of gene expression and splicing, we compared predicted gene expression and splicing for CMC DLFPC samples to measured ROSMAP DLFPC gene expression and splicing. We found a highly significant replication (mean $R^2$ for expression $= 0.071$; $p = 1.3 \times 10^{-36}$; mean $R^2$ for splicing $= 0.047$; $p = 2.1 \times 10^{-29}$), with 50.2% genes and splicing having $R^2 > 0.01$ (Supplementary Figure 9). Together, these results suggest that fitted models in CMC predict similar levels of *cis*-regulated expression on average in ROSMAP.

We then attempted to validate the TWAS results from models fitted in CMC with models fitted in ROSMAP. We found that 21 and 11 out of 44 genes replicated at a nominal $p < 0.05$ and an adjusted $p < 0.001$ (0.05/44), respectively, with expression and splicing models from ROSMAP. Interestingly, 8 out of 44 TWAS genes located in PD GWAS suggestive regions ($5 \times 10^{-8} < p < 1 \times 10^{-6}$) were replicated in ROSMAP. The direction of effect for most of the associations was concordant (Supplementary Table 1; Supplementary Figure 10). Thus, the genetic effects on splicing

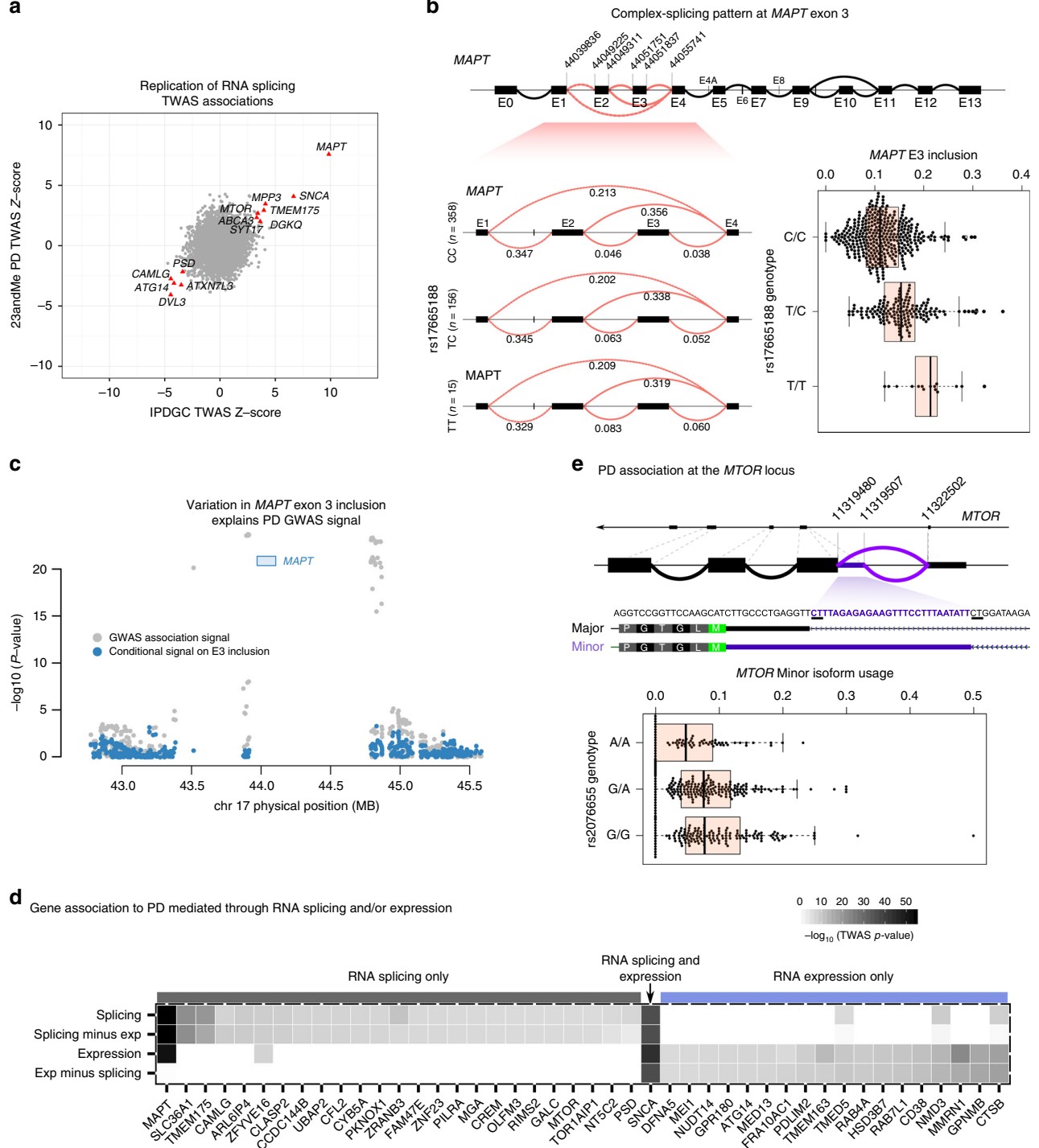

**Fig. 3** Parkinson's disease (PD) risk alleles affect splicing of nearby genes. **a** Replication of transcriptome-wide association study (TWAS) Z-score across two PD cohorts (23andMe and IPDGC). **b** TWAS signal at the *MAPT* locus is explained by the single-nucleotide polymorphisms (SNP) rs17665188, which is associated with *MAPT* exon 3 inclusion levels. rs17665188 tags the known H1/H2 haplotypes, which have been shown to be associated with PD[45]. The T allele, which tags the H2 haplotype, doubles inclusion of exon 3 and is associated with increased risk for PD. **c** PD TWAS signal at the MAPT locus (gray) and TWAS signal after removing the effect of *MAPT* exon 3 inclusion (cyan). This analysis shows that the association is largely explained by *MAPT* exon 3 inclusion. **d** Heatmap of genes identified from our TWAS analysis using imputed RNA splicing and expression (splicing and expression, respectively). The rows "splicing minus exp" and "exp minus splicing" denote association strengths after conditioning on expression and splicing, respectively. *SNCA* is the only gene associated with PD through both a genetic effect on RNA splicing and expression. **e** PD-associated variants at the *MTOR* locus are associated with an increase in a minor *MTOR* isoform that extends the 5′ of an exon in the 5′-untranslated region (5′-UTR)

are robust and unlikely to be due to artifacts unique to one of the datasets. To further replicate our findings, we repeated our TWAS using the CMC data and the PD GWAS summary statistics[31] (using the 23andMe cohort only). When we compared the Z-scores obtained from these two TWAS, we found that 12 genes discovered in International Parkinson Disease Genomics Consortium (IPDGC) cohort (e.g., *MAPT, SNCA, ATG14, DVL3,* and *MTOR*) could be replicated (Fig. 3a). These two complementary replication efforts demonstrate the robustness of our TWAS results.

**PD genetic variants often affect splicing of nearby genes.** To obtain a better understanding of the associations identified in our TWAS, we next focused on dissecting the mechanisms by which specific genes were associated with PD. We identified many known PD genes, such as *MAPT*, the microtubule-associated protein *tau* (chr17:44049311:44055741; $p = 2.60 \times 10^{-24}$, Fig. 3b) and *SNCA* (chr4:90756843:90757894; $p = 1.70 \times 10^{-16}$) that were associated with PD through RNA splicing. However, several genes, including *MAPT*, were found in both our RNA expression and splicing TWAS analyses. This finding is consistent with the existence of contradictory studies that either proposes that differences in *MAPT* expression[8], or that differences in *MAPT* splicing (exon 3 inclusion[44] or exon 10 inclusion[8]) are driving the PD genetic association at this locus. We, therefore, asked whether the associations we identified were more likely to be mediated through RNA splicing or through RNA expression and whether we could predict which gene or splicing event is most likely to be causal. At the *MAPT* locus, six associations to splicing events in three genes were identified. Using FUSION[14], we found that only one splicing event (inclusion of *MAPT* exon 3) remained significant after conditioning on all other association signals, which include *MAPT* total RNA expression levels and other *MAPT* RNA splicing events (conditional $p < 2.60 \times 10^{-24}$; Fig. 3c, d; Supplementary Table 2). To identify which genetic variant is associated to *MAPT* exon 3 inclusion, we searched our splicing QTL data from DLPFC and found that a nearby SNP (rs17665188/chr17:44357351) was strongly associated to exon 3 inclusion levels. Importantly, this SNP tags ($r^2 > 0.93$) the two well-known haplotypes (H1/H2)[44]. Haplotype H2 is associated with increased *MAPT* exon 3 inclusion in DLPFC ($p < 2.2 \times 10^{-16}$, LR *t*-test). Therefore, we conclude that splicing variation of *MAPT* exon 3 in the brain explains the reported association between the H1/H2 haplotypes and PD[45].

While some of the genes we identified were located within loci previously identified using GWAS, we discovered 47 associations that were in PD GWAS loci at suggestive level of significance (Supplementary File 1). Most of these were located in loci with suggestive associations in PD GWAS ($5 \times 10^{-8} < p < 1 \times 10^{-6}$, Supplementary Figures 11–49), and four genes (*CTSB, PDLIM2, GALC,* and *C8orf5*) were previously found to be genome-wide significant in PD GWAS meta-analysis[3]. One of these genes is cathepsin B (*CTSB*), which is a part of protease essential in α-syn lysosomal degradation[46]. We also detected *MTOR* as a novel candidate PD gene (Fig. 3e; Supplementary Figure 11). *MTOR* is a highly conserved serine/threonine kinase expressed in most mammalian cell types and plays a central role in the regulation of autophagy[47,48]. Of note, recent data have shown that dysregulation of mammalian target of rapamycin (mTOR) is implicated in the pathogenesis of PD[49,50], and it has been suggested as a novel therapeutic target for PD. Here, we found that a putative PD risk SNP (rs207655) was associated with a significant increase in usage of a rare *MTOR* isoform (Fig. 3e). Our results highlight the important role of RNA splicing in mediating the effects of risk loci on PD.

**PD TWAS genes form coherent functional pathways.** We hypothesized that the newly identified TWAS genes may be part of the same network or pathway as known PD susceptibility genes. To test this, we used GeNets[51] to measure the protein-protein interaction (PPI) network connectivity of our TWAS genes with known PD susceptibility genes. Most of these known PD susceptibility genes (*PARK2, PARK7, PINK1*) were linked to PD through studies on familial forms of PD and do not harbor functional common genetic variation. Thus, GWAS and TWAS approaches will not be able to identify these known PD susceptibility genes. Nevertheless, we reasoned that putative PD genes identified using our TWAS approach may functionally interact with these genes through protein-protein or gene regulatory interactions.

As expected, we found that monogenic PD genes form a PPI network that is directly connected (i.e., they form shared communities) to genes in PD GWAS loci ($p < 7.2 \times 10^{-3}$) (Fig. 4a). When we incorporated the TWAS genes with known PD susceptibility genes (including the monogenic genes), we found an expanded PPI network with four distinct communities of genes encoding for proteins that physically interact (Fig. 4b; $p < 2.3 \times 10^{-3}$). The genes in the TWAS-prioritized PPI network are highly enriched for biological pathways including lysosome (Bonferroni $p = 0.0017$) and α-syn aggregation pathways (Bonferroni $p = 1.8 \times 10^{-5}$) (Fig. 4b). These results support our hypothesis that the novel candidate PD genes we identified in this study are part of a larger set of interacting genes with coherent biological function, of which the lysosome pathway may be particularly central in the etiology of PD.

**Discussion**
In this study, we demonstrate the ability of our TWAS approach to detect a large number of genes relevant to PD without any prior information. This suggests that our approach may be widely applicable to other complex traits and diseases. Importantly, our work highlights RNA splicing as an important mediator of genetic effects on disease and therefore implies that future TWAS should include RNA splicing as an intermediate molecular phenotype when large-scale RNA-seq data is available in a relevant cell type or tissue.

Our work also advances our understanding of PD in a genomic context. For example, while a growing body of evidence has implicated several mechanisms such as innate-immune response in PD pathophysiology, it remains unclear which genes and specific pathways are involved. Our analysis of gene expression in monocytes suggests that PD-associated genetic risk factors influence innate-immune mechanisms. Although recent work suggests that *LRRK2* levels are increased in monocytes of PD patients[52], a causal relationship between PD susceptibility and *LRRK2* was not established. By combining PD GWAS and gene expression data we provide evidence that the common variant rs76904798 regulates the expression of *LRRK2* in peripheral monocytes but not in the cortex (Fig. 2b). Further experimental work is necessary to understand the mechanisms by which *LRRK2* expression may modulate monocyte gene expression and function in PD.

We also leveraged PD GWAS data and large-scale transcriptomic data from human cortex to identify genes for which the genetic component of expression level or differential splicing is associated with PD. TWAS corroborated many of the known PD genes but also identified several candidate disease genes in suggestive PD loci. Our analysis implicates gene expression and splicing regulation in cortical tissue as key mechanisms that mediate genetic risk for PD. Notably, genes in suggestive PD loci (e.g., *CTSB* and *MTOR*) identified in our study, but not in PD

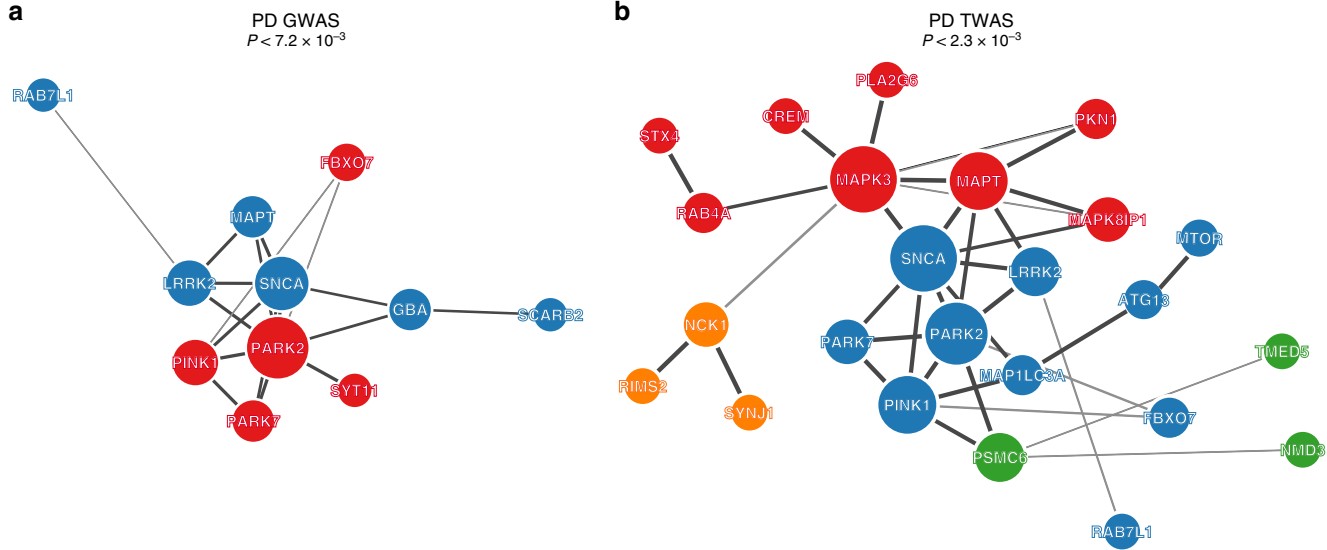

**Fig. 4** Proteins nominated by Parkinson's disease transcriptome-wide association study (PD TWAS) form expanded protein-protein interaction (PPI) and are enriched in the lysosomal pathway. **a** The PPI network connectivity are statistically significant ($p < 7.2 \times 10^{-3}$) and form two communities (represented by blue and red nodes). **b** We also applied GeNets to evaluate PPI network connectivity between protein products nominated by PD TWAS and monogenic genes. We found that they form a significantly expanded PPI ($p < 2.3 \times 10^{-3}$) network with known PD genes with four communities (represented by blue, green, orange, and red nodes)

GWAS, further support growing evidence of the involvement of lysosomal functions in the etiology of PD.

Our study prioritizes genes for subsequent experimental follow-up, which will help interrogate the molecular mechanisms underlying PD. Our catalog of splicing QTLs in DLFPC are made available with this study (see URLs) and provides a starting point for further mechanistic work to elucidate the role of associated genes in PD. More importantly, our approach is widely applicable to complex traits for which GWAS data are available. We expect that TWAS using both RNA expression and RNA splicing as intermediates will be a powerful strategy to prioritize disease-relevant genes, particularly in light of the increasing number of large-scale transcriptome datasets, such as the ones from the GTEx Consortium, that are publicly available.

## Methods

**Transcriptomic atlases for LDSC-SEG**. Data for LDSC-SEG was prepared as described in ref. [22]. The following data were used in our LDSC-SEG analyses: (1) GTEx samples that had more than four samples with at least one read count per million and at least 100 genes with at least one read count per million were TPM normalized[53]. (2) Publicly available data for DEPICT[17] was downloaded and pruned so that no two tissues had $r^2 > 0.99$. Two sets of highly correlated tissues (eyelid, conjunctiva, anterior eye segment, tarsal bones, foot bones, and bones of the lower extremity; and connective tissue, bone and bones, skeleton, and bone marrow) were removed completely, leaving 152 tissues. (3) Affymetrix–; GeneChip expression array data from mouse forebrain sorted cells were downloaded from GEO (GSE9566)[23]. (4) Publicly available gene expression data from the ImmGen project[18] was downloaded from GEO (GSE15907, GSE37448). Ensembl orthologs were used to map to human genes.

**Transcriptomics panels for TWAS**. The following panels were used in our study: (1) CMC RNA-seq data: generation and processing were previously described[21]. Briefly, DLPFC (Brodmann areas 9/46) was dissected from post-mortem brains of 258 individuals with schizophrenia and 279 control subjects. These individuals were of diverse ancestry, had no AD or PD neuropathology, had no acute neurological insults (anoxia, stroke, or traumatic brain injury) immediately before death, and were not on ventilators near the time of death. Total RNA was isolated from homogenized tissue and ribosomal RNA depleted. One hundred base pair paired-end reads were obtained using an Illumina–; HiSeq 2500, and mapped using TopHat. Genotyping was performed using the Illumina Infinium HumanOmniExpressExome-8 v1.1b chip. (2) Fairfax et al. monocyte expression data generation was previously described in ref. [19]. In short, blood was collected from 432 individuals of European ancestry, and CD14$^+$ monocytes were isolated from peripheral blood mononuclear cells via magnetic activated cell sorter. RNA

was quantified using the Illumina HumanHT-12 v4 BeadChip. Expression was normalized and transformed using robust spline normalization in R using the Lumi package, and corrected for batch effects using the ComBat package. The effect of incubation time on expression was regressed out of the normalized expression values. Genotyping was performed using the Illumina HumanOmniExpress-12 v1.0 chip. (3) Cardiogenic monocyte expression data generation was previously described in[20,54]. Monocytes were sorted from whole blood from individuals of European descent, and expression was assessed from RNA using the Illumina HumanRef 8 v3 Beadchip. Individuals were genotyped on Illumina Human 610 Quad custom arrays. The gene expression data is available via the European Genome-phenome Archive (EGA ID: EGAS00001000411). (4) ROSMAP RNA-seq data generation was previously described in refs. [27,28]. ROSMAP is a prospective cohort of aging individuals, where individuals are healthy at enrollment and 38% have clinical Alzheimer's disease at the time of death. DLPFC gray matter was dissected from 540 post-mortem brains. Sequencing libraries were prepared using a strand-specific dUTP protocol with poly-A selection, and sequencing on an Illumina HiSeq produced 101 bp paired-end reads. Individuals were genotyped on an Affymetrix GeneChip 6.0. For all transcriptome datasets used in this study see Supplementary Table 3.

**GWAS datasets**. We performed TWAS using PD GWAS summary statistics from Nalls et al.[31]. For discovery, we restricted our GWAS from PD cases and controls to International Parkinson Disease Genomics Consortium (IPDGC), PD GWAS Consortium, The Cohorts for Heart and Aging Research in Genomic Epidemiology (CHARGE) Consortium, PDGENE, and Ashkenazi studies cohorts (9581 cases and 33245 controls). For replication, we used the summary statistics from 23andMe subsets only (v2 and v3) from Nalls et al.[31] (4124 cases and 62,037 controls).

**TWAS studies**. TWAS is a powerful strategy that integrates SNP-expression correlation (*cis*-SNP effect sizes), GWAS summary statistics and LD reference panels to assess the association between the *cis*-genetic component of expression and GWAS[14]. TWAS can leverage large-scale RNA-seq data to impute tissue-specific genetic expression levels from genotypes (or summary statistics) in larger samples, which can be tested to identify potentially novel associated genes[13,14]. We used the FUSION tool[55] (see URLs) to perform TWAS for each transcriptome reference panel. The first step in FUSION is to estimate the heritability of each feature (gene expression or intron usage) using a robust version of GCTA-GREML[33], which generates heritability estimates per feature as well as the likelihood ratio tests $p$ value. Only genes or intron usage that were significant for heritability estimates at a Bonferroni-corrected $p < 0.05$ were retained for further analysis. The expression or intron usage predictive weights were computed by five different models implemented in the FUSION framework: best linear unbiased prediction, Bayesian sparse linear mixed model, LASSO, Elastic Net, and top SNPs. A cross-validation for each of the desired models are then performed. The model with the best cross-validation prediction accuracy are used for predicting expression or intron usage into the GWAS. The imputed gene expression or intron usage are then used to correlate to PD GWAS summary statistics (see "GWAS Datasets")

to perform TWAS and identify significant associations. To account for multiple hypotheses, we applied an FDR of 5% within each expression and splicing reference panel (see "Transcriptomics panels for TWAS") that was used.

**Joint and conditional analysis**. Joint and conditional analyses of each locus with multiple TWAS association signal were performed using the summary statistic-based method described in refs. [55,56]. This approach requires TWAS association statistics and a correlation matrix to evaluate the joint/conditional model. The correlation matrix was estimated by predicting the *cis*-genetic component of expression (or intron usage) for each TWAS gene/intron cluster into the 1000 Genomes genotypes. Then, Pearson's correlations were calculated across all pairs of genes/intron cluster and between all gene/SNP pairs. The FUSION tool (see URLs) was used to perform the joint and conditional analyses and generate regional scatterplots.

**Splicing QTL mapping**. We used Leafcutter[32] to obtain intron excision ratios, which is the proportion of intron defining reads to the total number of reads from the intron cluster it belongs to. We used the alignments from STAR as an input to LeafCutter. Before quantifying intron excision ratios, we used WASP[57] to remove read-mapping biases caused by allele-specific reads. The intron excision ratios were standardized across individuals for each intron and quantile normalized across introns[15]. The normalized intron excision ratios were used as our phenotype matrix. To map sQTLs, we used linear regression (implemented in fastQTL[58]) to test for associations between SNP dosages (minor allele frequency (MAF) > 0.01) within 100 kb of intron clusters and the rows of our phenotype matrix that correspond to the intron excision ratio within each intron cluster. We used the first three principal components of the genotype matrix to account for the effect of ancestry plus the first 15 principal components of the phenotype matrix to regress out the effect of known and hidden confounding factors. An adaptive permutation scheme[58] (implemented in fastQTL) was used to estimate the number of sQTLs for any given FDR. An empirical *p* value for the most significant QTL for each intron cluster was calculated. Benjamini–Hochberg correction on the permutation *p* values was applied to extract all significant sQTL pairs with an FDR of 5%. All significant sQTLs are available via the DLPFC sQTL browser (see URLs).

**Colocalization**. We used coloc 2.3-1[43] to colocalize the PD association signal at TWAS loci with QTL signals. For each locus, we examined all SNPs available in both datasets within 500 Mb of the SNP identified in TWAS as the top QTL SNP, and ran coloc.abf with default parameters and priors. We called the signals colocalized when (coloc H3 + H4 ≥ 0.8 and H4/H3 ≥ 2)[58].

**GWAS enrichment**. The GARFIELD tool was used to test for enrichment of GWAS SNPs among sQTL and other publicly available QTL datasets[30]. GARFIELD annotates GWAS SNPs (LD pruned; $r^2 > 0.1$) based on functional information overlap. Then, quantifies fold enrichment at GWAS $p < 10^{-5}$ cut-off and assesses the significance by permutation testing (matching the SNP sets for MAF, distance to nearest transcription start site, and the number of LD proxies).

**PPI network and pathway analysis**. The GeNets online tool (see URLs) was used to construct the PPI networks. GeNets builds PPI network using evidence of physical interaction from the InWeb database, which contains 420,000 high-confidence pair-wise interactions involving 12,793 proteins[51]. GeNets displays community structures, which are also known as modules or clusters. The community structures highlight genes that are more connected to one another than they are to other genes in other modules. To assess the statistical significance of the networks, GeNets applies a within-degree node-label permutation strategy. Briefly, it builds random networks that mimic the structure of the original network and evaluates network connectivity parameters on these random networks to generate empirical distributions for comparison to the original network. In addition to PPI network construction, GeNets allows for gene set enrichment analysis on genes within the PPI network. The following gene sets and databases were used for enrichment analysis: Molecular Signatures Database (MSigDB) Curated Gene Sets (C2), Kyoto Encyclopedia of Genes and Genomes (KEGG), BioCarta, and Reactome. A hypergeometric testing is applied to get *p* value for gene set enrichment.

**URLs**. For FUSION, see http://gusevlab.org/projects/fusion/; for DLPFC sQTL browser, see https://rajlab.shinyapps.io/sQTLviz_CMC/; for CMC Knowledge Portal, see https://www.synapse.org/#!Synapse:syn2759792; for AMP-AD Knowledge Portal, see https://www.synapse.org/#!Synapse:syn2580853; for Rush Alzheimer's Disease Center Research Resource Sharing Hub, see http://www.radc.rush.edu; for GeNets, see https://apps.broadinstitute.org/genets.

## Data availability
TWAS results, sQTL summary statistics, and splicing visualization browser is available at https://rajlab.shinyapps.io/sQTLviz_CMC/. FUSION software, CMC weights, and reference LD are available at http://gusevlab.org/projects/fusion/. RNA-seq data for CMC is available via the CMC Knowledge Portal, https://www.synapse.org/#!Synapse:

syn2759792. RNA-seq and genotype data for ROSMAP is available via the AMP-AD Knowledge Portal, https://www.synapse.org/#!Synapse:syn2580853. The precomputed DLFPC and monocytes gene expression or splicing weights are available upon request from the corresponding author.

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

## Acknowledgements

This work was supported by grants from the US National Institutes of Health (NIH grant R01AG054005) and the Michael J. Fox Foundation. This work was supported in part through the computational resources and staff expertise provided by Scientific Computing at the Icahn School of Medicine at Mount Sinai. We would like to thank Elisa Navarro, Hae Kyung Im, and Tim Ahfeldt for their insightful comments on the paper. We thank the patients and families who donated material for CMC data. The CommonMind Consortium data are available in CMC Knowledge Portal (see URLs). Data were generated as part of the CMC supported by funding from Takeda Pharmaceuticals Company Limited, F. Hoffman-La Roche Ltd and NIH grants R01MH085542, R01MH093725, P50MH066392, P50MH080405, R01MH097276, RO1-MH-075916, P50M096891, P50MH084053S1, R37MH057881 and R37MH057881S1, HHSN271201300031C, AG02219, AG05138, and MH06692. Brain tissue for the study was obtained from the following brain bank collections: the Mount Sinai NIH Brain and Tissue Repository, the University of Pennsylvania Alzheimer's Disease Core Center, the University of Pittsburgh NeuroBioBank and Brain and Tissue Repositories and the NIMH Human Brain Collection Core. CMC Leadership: Pamela Sklar, Joseph Buxbaum (Icahn School of Medicine at Mount Sinai), Bernie Devlin, David Lewis (University of Pittsburgh), Raquel Gur, Chang-Gyu Hahn (University of Pennsylvania), Keisuke Hirai, Hiroyoshi Toyoshiba (Takeda Pharmaceuticals Company Limited), Enrico Domenici, Laurent Essioux (F. Hoffman-La Roche Ltd), Lara Mangravite, Mette Peters (Sage Bionetworks), Thomas Lehner, Barbara Lipska (NIMH). We thank the participants of ROS and MAP for their essential contributions and gift to these projects. The ROSMAP data are available at the Rush Alzheimer's Disease Center (RADC) Research Resource Sharing Hub (see URLs). The ROSMAP and MSBB mapped RNA-seq data that support the findings of this study are available in AMP-AD Knowledge Portal (see URLs) upon authentication by the Consortium. This work has been supported by many different NIH grants: P30AG10161, U01AG046152, R01AG16042, R01AG036836, R01AG015819, R01AG017917, and R01AG036547. Lastly, we thank the research participants and employees of 23andMe who contributed to the PD GWAS in ref. [31].

## Author contributions

T.R. conceived of the project. Y.I.L., G.W., J.H., and T.R. performed the analyses. T.R. and Y.I.L. wrote the manuscript.

## Additional information

**Competing interests:** The authors declare no competing interests.

