## [Peer Review File · Nature Communications]

Reviewer #1 (Remarks to the Author):

Prioritizing Parkinson's Disease genes using population-scale transcriptomic data

=====

Overview

The manuscript describes a TWAS in Parkinson's disease (PD) using predicted total and splicing variation in brain- and

immune-cell types using publicly available PD GWAS data. The authors report finding 44 risk genes for PD and that

results are enriched in known pathways for PD risk. The manuscript is written clearly. However I have comments regarding

inconsistency in presentation of results and lack of crucial details for predictive accuracy, testing details, and

analyses.

Main comments

1) Tissue Enrichment Analysis

1.1) The authors perform an integrative analysis using a bioinformatic tool LD-SEG to discern which tissues are likely

most relevant for PD risk. Their results modestly suggest CNS has the largest contribution when using GTEx data.

The authors then argue that monocytes are more important for AD when compared with PD. This argument, while

scientifically interesting it seems out-of-place in the context of the broader manuscript, considering this is the only

result comparing the two traits.

2) TWAS

2.1) The authors describe building predictive models for downstream TWAS using monocytes and DLFCP gathered across

several data sources yet provide no details on how accurate the fitted models are. Can the authors provide statistics

across models (e.g., total number of models, heritability, out-of-sample R², which linear model was selected most often,

R² / h²g)? These statistics are critical in providing context for how models will perform in TWAS. As currently written,

it isn't clear what defines significance in monocyte PD-TWAS. I'm assuming its 0.05 / #-of-models, but this isn't

stated.

2.2) The manuscript states 'FDR at 5%' was used to determine significance for DLFP. I'm assuming now that this is the

same criterion for the monocyte TWAS.

2.3) The authors describe a nice colocalization approach to provide additional evidence of eQTLs mediating risk for PD,

yet do not describe similar results in the monocytes section.

2.4) There is striking lack of colocalization between sQTLs in DLFP and PD, despite earlier analysis reporting strong

enrichment. The authors claim this can be explained by "smaller" sQTL -> PD effects. The authors provide a very helpful

coloc figure describing the density of posterior estimates. Another simple approach would be to measure the average TWAS

chi-squared statistic across sQTL and eQTL models. When expression (splicing) mediates risk for PD, the squared test

statistic will be proportional to the squared normalized effect sizes.

2.5) The authors provide no high-level summary of the GWAS. How many GWAS risk regions are explained by TWAS?

2.6) The authors state that 12/44 ROS/MAP genes replicated at a nominal significance threshold (P < 0.05) which partially

supports their claim. Reporting how many genes replicated at an adjusted P-value (P < 0.05 / 44) would strengthen their

claim.

2.7) The authors perform a validation in DFLPC for PD using expression from an independent study (ROS/MAP). Can the

authors describe how well fitted models replicate before describing TWAS replication? How many genes were heritable?

What is the average cross-study R^2 ? In other words, using fitted models in ROS/MAP what is the R^2 in measured expression

for CommonMind measurements (also adjusted for SNP heritability in common mind). This will provide critical quality

assessment before proceeding with TWAS replication.

2.8) A validation using independent(?) GWAS for PD was performed with 12 genes replicated. It would be helpful if the

authors provide more details about what defines a replication. Similarly, what is the probability of observing the

overlapping gene-set by chance?

2.9) Overall I think the manuscript would be significantly strengthened by first summarizing predictive models and TWAS results across all tissues and datatypes. This will help the reader get a broader picture of prediction accuracy, risk regions identified, genes identified, and GWAS regions explained. It would then set the tone for dissecting tissue-specific results.

3) PD risk variants -> splicing

3.1) This section describes a very nice breakdown for testing MAPT splicing for association with PD risk. A conditional

analysis is performed but the language used is slightly imprecise. Can the authors clarify the statement,

"after conditioning on all other association signals, including MAPT total RNA expression levels."

Does this mean all

association signals for MAPT (significant -and- non-significant) at this region? Or all tested gene models MAPT, or

otherwise?

3.2) The authors leverage TWAS's lower testing burden to detect "suggestive" risk regions (i.e. TWAS-significant, but

GWAS-suggestive). It would be illustrative to describe how many of the reported suggestive regions replicated in the

independent study. TWAS as a tool is useful to provide mechanistic hypotheses at -known- risk regions, however at

regions with less SNP-based evidence, independent results are highly informative.

4) PPI + Pathways

4.1) The authors report significant results for enrichment of lysosome function to bolster claims of TWAS genes involved in lysosome function and mediating downstream PD risk. What is not clear is whether pathway analysis

was performed on the set of TWAS genes only, or the PPI network that consisted of TWAS genes with the addition of known

PD susceptibility genes. It would be very helpful if the authors could clarify this, and if necessary, recompute pathway

analysis using TWAS-significant only genes.

Minor comments

1) Figure 1c lists the chart as 'sQTLs among PD GWAS hits'; yet the text describes eQTLs, mQTLs, etc.

Reviewer #3 (Remarks to the Author):

The authors present a transcriptome wide association study in Parkinson's disease (PD). Existing GWAS data and imputed RNA expression and splicing data had been jointly analyzed by bioinformatics approaches. This study pursues an innovative approach to identify proteins implicated in the pathophysiology of PS.

The authors should consider the following major remarks:

(1) Nature Communications is not bioinformatics specialist journal, but a general science journal. In large parts, the current text reads like insider slang, which is probably only comprehensible very few working groups in the world. Most non-standard abbreviations are not defined appropriately (e.g. DLPFC, LDSC-SEG, sQTLs, GARFIELD). Most figure legends are not

comprehensible by themselves without consulting the results and methods part and looking up the original texts of the cited literature. No methodological details, such as case numbers, are reported adequately. Even the information of the primary source of their datasets is well hidden at the end of the manuscript rather than presented at the first occurrence. In Fig 1b, the authors report Alzheimer's disease data without providing any information in the manuscript on which original data this analysis is based. In the current version, the manuscript is incomprehensible to the vast majority of Nature Communications readers.

(2) Some information claimed in the manuscript is not represented in the corresponding figures, e.g. page 3, line 7 (amygdala, substantia nigra, anterior cingulate cortex, frontal cortex, hypothalamus, and cervical (C1) spinal cord) is not shown in Fig. 1b, as indicated in the text.

(3) A critical limitation is that the GWAS data were used on the level of summary association statistics (e.g. individual SNP effect sizes), rather than raw data. Please specify your imputation methodology in detail (imputation of expression-trait association statistics or imputation of individual-level data). Please reanalyze at least the publicly available subset of original GWAS genotyping raw data.

(4) The validity of the TWAS predictions has to be verified in the transcriptome of PD vs. control brains. Please verify if your predicted transcripts in prior transcriptomic profiling studies of tissues from PD vs. controls (e.g. Mariani et al., PLoS One. 2016 Sep 9;11(9):e0161567. Mariani et al., Genes (Basel). 2018 May 25;9(6)). Ideally, verification of predicted expression differences between PD and controls should be verified on transcript and protein level in postmortem brains of PD and controls (sufficiently powered case number).

(5) The largest PD GWAS (Chang et al. 2017) compared 6,476 PD cases with 302,042 controls, followed by a meta-analysis with over 13,000 PD cases and 95,000 controls. The authors however used an older GWAS dataset with lower case numbers (Nalls et al., 2014). Why?

(6) The Z-score for SNCA in PD association with RNA expression in brains and Z-score for SNCA PS association with peripheral monocyte RNA expression point in opposite directions, suggesting reduced alpha Synuclein expression in brains of PD patients. This is completely at odds with our current concept of PD pathophysiology. Please explain.

(7) The authors use the SNP rs17665188 for the H1/H2 polymorphism of MAPT. Nalls et al., however reported rs17649553 as primary PD-associated MAPT variant. Why did you emphasize a different SNP?

(8) The authors conclude that genes in suggestive PD loci (e.g., CTSB) were identified in their study, but not in PD GWAS before. This is not correct, since CTSB had already been reported in the Chang 2017 PD GWAS.

Authors' point-by-point response to the reviewers

Title: Prioritizing Parkinson's Disease genes using population-scale transcriptomic data

Manuscript reference number: *Nature Communications*, Li et al. 2018

Response to the Reviewers:

We are grateful to the Editor for inviting us to submit a revised version of the manuscript. We thank the two reviewers for critical feedback, thoughtful recommendations, and support of this work. The reviewers' found that this work of "considerable potential interest...", but provided several suggestions to strengthen our manuscript. To address a major concern from reviewer 1, we have now considerably restructured our manuscript. We have also added details throughout our manuscript to make our analyses and results more easily interpretable. All changes are highlighted in the manuscript text file. The response to reviewers comments are in blue font.

Reviewers' comments:

Reviewer #1 (Remarks to the Author):

Prioritizing Parkinson's Disease genes using population-scale transcriptomic data

=====

Overview

The manuscript describes a TWAS in Parkinson's disease (PD) using predicted total and splicing variation in brain- and immune-cell types using publicly available PD GWAS data. The authors report finding 44 risk genes for PD and that results are enriched in known pathways for PD risk. The manuscript is written clearly. However I have comments regarding inconsistency in presentation of results and lack of crucial details for predictive accuracy, testing details, and analyses.

Main comments

1) Tissue Enrichment Analysis

1.1) The authors perform an integrative analysis using a bioinformatic tool LD-SEG to discern which tissues are likely most relevant for PD risk. Their results modestly suggest CNS has the largest contribution when using GTEx data. The authors then argue that monocytes are more important for AD when compared with PD. This argument, while scientifically interesting it seems out-of-place in the context of the broader manuscript, considering this is the only result comparing the two traits.

We thank the reviewer for this point. Our goal here is to have a disease as a comparison group with PD and to validate the LD-SEG results. We did not intend to directly compare the cell type enrichment of the two diseases. In terms of the rest of the manuscript, we did compare TWAS results between the two diseases. In particular, we highlight several loci that are shared between the two diseases (*CD33* and *PILRA*). While we think that our results suggest that monocytes are more important players in the development of AD when compared with PD, our results suggests several genes expressed in monocytes may play a role in PD when specific functions (e.g. lysosomal) are considered. We have modified the main text to clarify some of the out-of-place texts (pg.3, paragraph 1).

2) TWAS

2.1) The authors describe building predictive models for downstream TWAS using monocytes and DLFPC gathered across several data sources yet provide no details on how accurate the fitted models are. Can the authors provide statistics across models (e.g., total number of models, heritability, out-of-sample R^2 , which linear model was selected most often, R^2 / h^2g)? These statistics are critical in providing context for how models will perform in TWAS. As currently written, it isn't clear what defines significance in monocyte PD-TWAS. I'm assuming its 0.05 / #-of-models, but this isn't stated.

We apologize for not including the TWAS statistics in the initial version of the manuscript. We agree with the reviewer that these statistics are critical in providing context for how models will perform in TWAS. Therefore, we now provide the complete statistics across models including number of models, heritability, model R^2 , linear model selected and r^2/h^2g as Supplementary Table 1.

We have made the following changes to the manuscript:

(1) Added a paragraph (main text, paragraph, page 4) to discuss the statistics across the predictive models:

“We used the FUSION software (see URLs) to estimate the heritability, to build predictive models, and to perform TWAS. For each reference panel, FUSION estimates the heritability of gene expression and alternative splicing explained by local SNPs (i.e., 1 Mb from TSS of each gene) using linear-mixed models. Genes or splicing events that meet are nominally significant at $P < 0.01$ for SNP-heritability (cis- h^2g) are then used for training predictive models. FUSION fits four predictive linear models (see methods) or every gene or intronic excision using local SNPs. The models with the best cross-validation prediction accuracy are kept for prediction into GWAS (Supplementary Fig. 3). In total, 17,798 tissue-specific models including 4,721 monocyte expression and 5,383 and 7,695 DLFPC total expression and alternatively spliced introns, respectively, corresponding to 8,648 unique genes were used for TWAS. The square of correlation (R^2) between predicted and observed gene expression levels normalized by corresponding cis- h^2g was calculated to measure prediction accuracy. Across all cohorts lasso attained the best predictive performance, with 30 % improvement in prediction R^2 over other models (Supplementary Figs. 4 & 5). We found that the normalized in-sample prediction accuracy (R^2 / cis- h^2g) to be 54%, 72% and 73% for monocytes, DLFPC-expression and DLFPC-splicing, respectively. These

result suggests that most of the signal in cis-regulated total expression and splicing levels is captured by the fitted models.”

(2) Added Supplementary Figures 3-5 and Supplementary Table 1 to report the statistics across models (e.g., total number of models, heritability, out-of-sample R², which linear model was selected most often, R² / h²g).

(3) Added a Supplementary Table 1 to include all the TWAS statistics for the significant results.

2.2) The manuscript states 'FDR at 5%' was used to determine significance for DLFPC. I'm assuming now that this is the same criterion for the monocyte TWAS.

Correct, a FDR 0.05 significance threshold was used for the monocyte PD-TWAS. We apologize for not stating this clearly. This is clarified in the main text (pg. 5; paragraph 2).

2.3) The authors describe a nice colocalization approach to provide additional evidence of eQTLs mediating risk for PD, yet do not describe similar results in the monocytes section.

The monocyte colocalization results was included in Supplementary Table S1. However, we now added the colocalization results from monocytes to main Figure 2. 10/29 monocyte eQTL colocalize with PD GWAS loci (Supplementary Table 1). We have also added a sentence in the main text to describe these numbers (pg. 5, paragraph 4)

2.4) There is striking lack of colocalization between sQTLs in DLFPC and PD, despite earlier analysis reporting strong enrichment. The authors claim this can be explained by "smaller" sQTL -> PD effects. The authors provide a very helpful coloc figure describing the density of posterior estimates. Another simple approach would be to measure the average TWAS chi-squared statistic across sQTL and eQTL models. When expression (splicing) mediates risk for PD, the squared test statistic will be proportional to the squared normalized effect sizes.

We found that the average TWAS chi square for expression and splicing were 20.46 and 19.22, respectively. However, it is not entirely clear to us how we can interpret these numbers as suggested by the reviewer.

2.5) The authors provide no high-level summary of the GWAS. How many GWAS risk regions are explained by TWAS?

We have now included this in Supplementary Table 1. **19 out of 41** of the PD GWAS risk regions are explained by TWAS (Supplementary Table 1). We added a sentence in the main text to include the details.

2.6) The authors state that 12/44 ROS/MAP genes replicated at a nominal significance threshold ($P < 0.05$) which partially supports their claim. Reporting how many genes replicated at an adjusted P-value ($P < 0.05 / 44$) would strengthen their claim.

We agree that adjusted P-value ($P < 0.05 / 44$) would strengthen the replication claim. We found that 11/44 genes replicated when we used an adjusted P-value ($P < 0.05/44$). We have updated the manuscript to reflect the number (pg. 7, paragraph #2).

2.7) The authors perform a validation in DFLPC for PD using expression from an independent study (ROS/MAP). Can the authors describe how well fitted models replicate before describing TWAS replication? How many genes were heritable? What is the average cross-study R²? In other words, using fitted models in

ROS/MAP what is the R2 in measured expression for CommonMind measurements (also adjusted for SNP heritability in common mind). This will provide critical quality assessment before proceeding with TWAS replication.

We have now included these information in the revised manuscript (pg. 7 paragraph 1):

“Before validating the TWAS results in ROSMAP, we evaluated cross-cohort prediction of the genes and intronic splicing in CMC and ROSMAP. The average $\{it\ cis\}$ -heritability estimates between the ROSMAP and CMC were strongly correlated across genes and intronic splicing (Pearson $\rho = 0.54$). The prediction accuracy between the two cohorts was also strongly correlated (Pearson $\rho = 0.43$ and 0.16 for R^2 and R^2/h^2_g , respectively) (Supplementary Figure 7-9). The weaker correlation of the normalized prediction accuracy is likely due to differences in the average $\{it\ cis\}$ -heritability estimates between ROSMAP ($\{it\ cis\}$ - $h^2_g = 0.098$) and CMC ($\{it\ cis\}$ - $h^2_g = 0.078$). To test the predictive consistency for models of gene expression and splicing, we compared predicted gene expression and splicing for CMC DLFPC samples to measured ROSMAP DLFPC gene expression and splicing. We found a highly significant replication (mean R^2 for expression = 0.071 ; $P = 1.3 \times 10^{-36}$; mean R^2 for splicing = 0.047 ; $P = 2.1 \times 10^{-29}$), with 50.2% genes and splicing have $R^2 > 0.01$ (Supplementary Figure 9). Together, these results suggest that fitted models in CMC predict similar levels of $\{it\ cis\}$ -regulated expression on average in ROSMAP.”

2.8) A validation using independent(?) GWAS for PD was performed with 12 genes replicated. It would be helpful if the authors provide more details about what defines a replication. Similarly, what is the probability of observing the overlapping gene-set by chance?

We consider a signal to replicate if the TWAS Z-scores are significant and in the same direction when using independent GWASs. 32 TWAS models were significant using an independent GWAS that consist of samples from the 23AndMe cohort (Nalls 2014) summary statistics and 66 when using the IPDGC dataset summary statistics at FDR 0.05. The probability of observing the overlapping gene-set by chance is extremely small (p-value = $p < 3.17e-13$).

2.9) Overall I think the manuscript would be significantly strengthened by first summarizing predictive models and TWAS results across all tissues and datatypes. This will help the reader get a broader picture of prediction accuracy, risk regions identified, genes identified, and gwas regions explained. It would then set the tone for dissecting tissue-specific results.

We thank the reviewer for this suggestion. We agree that summarizing the predictive models and TWAS results would significantly strengthened the manuscript. We have now reorganized the presentation of our results based on this reviewer’s comments. Specifically, we now start with presenting the TWAS predictive models and results across all tissues and datatypes.

We added a new paragraph to page 4 and 5:

“We used the FUSION software (see URLs) to estimate the heritability, build predictive models, and perform TWAS. For each reference panel, FUSION estimates the heritability of gene expression and alternative splicing explained by local SNPs (i.e., 1 Mb from TSS of each gene) using linear-mixed models. Genes or splicing events that are nominally significant at $P < 0.01$ for SNP-heritability ($\{it\ cis\}$ - h^2_g) are used for training predictive models (Supplementary Fig.~3). FUSION fits four predictive linear models (see Methods) for every gene or intronic excision event using local SNPs as predictors. The models with the best cross-validation prediction accuracy are kept for prediction into GWAS (Supplementary Figs.~4 and 5). In total, 17,798 tissue-specific models including 4,721 monocyte expression, 5,383 DLPFC expression, and 7,695 DLFPC alternatively spliced introns, were used for TWAS. The square of correlation (R^2) between

predicted and observed gene expression levels normalized by corresponding cis-h2g was calculated to measure prediction accuracy. Across all cohorts LASSO attained the best predictive performance, with 30 % improvement in prediction R2 over other models (Supplementary Fig. 5). We found the average in-sample prediction accuracy ($R^2 / cis-h^2_g$) to be 54%, 72% and 73% for monocyte, DLFPC-expression and DLFPC-splicing, respectively. These results are consistent previous TWAS analyses (Gusev et al. 2016) and suggest that most of the signal in cis-regulated total expression and splicing levels is captured by the fitted models.”

3) PD risk variants -> splicing

3.1) This section describes a very nice breakdown for testing MAPT splicing for association with PD risk. A conditional analysis is performed but the language used is slightly imprecise. Can the authors clarify the statement, "after conditioning on all other association signals, including MAPT total RNA expression levels." Does this mean all association signals for MAPT (significant -and- non-significant) at this region? Or all tested gene models MAPT, or otherwise?

To condition our splicing signal on other signals, we used FUSION (Gusev et al. 2018 Nature Genetics) to test whether *MAPT* splicing events and *MAPT* total gene expression levels are marginally significant or jointly significant. We found that in the *MAPT* locus, only *MAPT* splicing and expression levels were significantly associated with PD. Thus, this means that while we tested splicing and expression of all other genes at the locus, the joint and conditional analyses were performed only for significant *MAPT* expression and RNA splicing association signals (also see Supplementary Figure S23). We have clarified this in the main text (pg. 7, paragraph 3).

3.2) The authors leverage TWAS's lower testing burden to detect "suggestive" risk regions (i.e. TWAS-significant, but GWAS-suggestive). It would be illustrative to describe how many of the reported suggestive regions replicated in the independent study. TWAS as a tool is useful to provide mechanistic hypotheses at -known- risk regions, however at regions with less SNP-based evidence, independent results are highly informative.

We think that one of the most interesting results from this study are the “suggestive” risk regions. We agree that TWAS is powerful approach to identify regions where there is less SNP-based evidence. We report the TWAS-significant but GWAS-suggestive results in Supplementary Table 1. Of the suggestive risk regions, 8 were significant in the independent study. Interestingly, using Nalls et al. 2014 summary statistics we prioritized *CTSB* as a candidate PD gene (with cis-eQTL effect in both brain and monocytes). The same gene was shown to reach genome-wide significance in the most recent GWAS (Chang et al. 2017).

4) PPI + Pathways

4.1) The authors report significant results for enrichment of lysosome function to bolster claims of TWAS genes involved in lysosome function and mediating downstream PD risk. What is not clear is whether pathway analysis was performed on the set of TWAS genes only, or the PPI network that consisted of TWAS genes with the addition of known PD susceptibility genes. It would be very helpful if the authors could clarify this, and if necessary, recompute pathway analysis using TWAS-significant only genes.

We performed the PPI and pathway analysis by using the set of TWAS genes plus known PD susceptibility genes. We compared the resulting network (TWAS+GWAS+ known genes) against the network build from the set of known PD susceptibility genes only. Our rationale for including known PD susceptibility genes is that many of these key PD genes do not harbor functional genetic variation, likely due to their large effect on PD

(e.g. PARK2, PARK7, PINK1). These known PD genes are found in monogenic and/or familial forms of PD and they generally cannot be found by GWAS or TWAS approaches as functional variants in these genes are likely under strong negative selection. Our analysis shows that the TWAS genes we have prioritized are not just set of random genes but some of them physically interact with the known PD genes, suggesting that they are functionally relevant. We have now added a clarification of our reasoning in the updated text.

Minor comments

- 1) Figure 1c lists the chart as 'sQTLs among PD GWAS hits'; yet the text describes eQTLs, mQTLs, etc.

We have now edited the caption in Figure 1c to emphasize that “sQTLs are enriched among PD GWAS hits”.

Reviewer #3 (Remarks to the Author):

The authors present a transcriptome wide association study in Parkinson's disease (PD). Existing GWAS data and imputed RNA expression and splicing data had been jointly analyzed by bioinformatics approaches. This study pursues **an innovative approach** to identify proteins implicated in the pathophysiology of PS.

The authors should consider the following major remarks:

(1) Nature Communications is not bioinformatics specialist journal, but a general science journal. In large parts, the current text reads like insider slang, which is probably only comprehensible very few working groups in the world. Most non-standard abbreviations are not defined appropriately (e.g. DLPFC, LDSC-SEG, sQTLs, GARFIELD). Most figure legends are not comprehensible by themselves without consulting the results and methods part and looking up the original texts of the cited literature. No methodological details, such as case numbers, are reported adequately. Even the information of the primary source of their datasets is well hidden at the end of the manuscript rather than presented at the first occurrence. In Fig 1b, the authors report Alzheimer's disease data without providing any information in the manuscript on which original data this analysis is based. In the current version, the manuscript is incomprehensible to the vast majority of Nature Communications readers.

We thank the reviewer for commenting on this and we have worked on making our manuscript more accessible to the audience of *Nature Communications*:

- (1) We made sure that all abbreviations are defined the first time they are mentioned, and also a second time if they appear significantly later than the first appearance.
- (2) We have updated all figure legends to define the non standard abbreviations.

(3) We have added the primary sources of dataset we used, where applicable and noted the sample sizes of the studies. A table that summarizes all the datasets with sample size, reference, tissues, etc used in this study are in Supplementary Table S3.

(2) Some information claimed in the manuscript is not represented in the corresponding figures, e.g. page 3, line 7 (amygdala, substantia nigra, anterior cingulate cortex, frontal cortex, hypothalamus, and cervical (C1) spinal cord) is not shown in Fig. 1b, as indicated in the text.

We are sorry if this was not clear in Figure 1b. These tissues are represented in red bars (Fig. 1b). These samples were the only GTEx tissues that were significantly enriched in PD heritability. We have now added text in the caption to describe the tissues from the GTEx project.

(3) A critical limitation is that the GWAS data were used on the level of summary association statistics (e.g. individual SNP effect sizes), rather than raw data. Please specify your imputation methodology in detail (imputation of expression-trait association statistics or imputation of individual-level data). Please reanalyze at least the publicly available subset of original GWAS genotyping raw data.

We thank the reviewer for this comment. One of the major limitation for many post-GWAS analysis such as ours is the limited availability of the raw genotype data. For this study, we had access to only the GWAS summary statistics due to limitation in accessing the 23andMe cohort raw data. Therefore, we were unfortunately unable to perform TWAS using individual level raw genotyping data.

The main contribution of our work is based on previously published imputation methods for TWAS. The imputation methodology details have been extensively documented in Gusev et al. 2016 Nature Genetics and Gusev et al. 2018 Nature Genetics. The authors have done excellent work in demonstrating that summary level and individual-level data produced highly consistent results. They did extensive comparison of summary-based TWAS and individual-level association Z-scores. The authors compared summary-based TWAS and individual-level TWAS of height using transcriptome reference from blood. The authors conclude that “summary-based TWAS is essentially identical to individual-level TWAS when using in-sample LD.” (Supplementary Figure 5 in Gusev et al. 2016 Nature Genetics). More recently, other studies have also reported similar results when comparing summary and individual level data with PrediXcan (another method similar to TWAS): “We find high concordance between PrediXcan and S-PrediXcan results indicating that in most cases, we can use the summary version without loss of power to detect associations.” (Barbeira et al. 2018 Nature Communications). Given these results, we are confident that our TWAS results are robust and that we have provided additional replication efforts in an independent transcriptome reference panels. We also acknowledge that we have provided a strategy to prioritize PD risk genes but there needs to be additional experimental validation to confirm our findings.

(4) The validity of the TWAS predictions has to be verified in the transcriptome of PD vs. control brains. Please verify if your predicted transcripts in prior transcriptomic profiling studies of tissues from PD vs. controls (e.g. Mariani et al., PLoS One. 2016 Sep 9;11(9):e0161567. Mariani et al., Genes (Basel). 2018 May 25;9(6)). Ideally, verification of predicted expression differences between PD and controls should be verified on transcript and protein level in postmortem brains of PD and controls (sufficiently powered case number).

We agree with the reviewer that our TWAS predictions has to be verified in the transcriptome of PD vs. control brains. However, at the moment, transcriptome profiles from PD vs. control brains are limited with the exception of few as provided by the reviewer and some ongoing projects in the corresponding authors lab. We note that these studies are underpowered ($n < 100$) to detect significant differential expression as samples from DLPFC are often highly heterogeneous, with many confounders to remove. Nevertheless, we have checked previously published results and highlighted the genes that are verified in previously published studies. Of the genes from our TWAS models, we found that *GPNMB* and *GBP7* were previously shown to be

differentially expressed in PD brains vs controls in Mariani et al., PLoS One. 2016. LRRK2 has also been shown to be differentially expressed in monocytes (Bliederhaeuser et al. 2016). We have updated the main text to include some of the verification of TWAS genes from published studies (pg. 5; paragraph 3).

(5) The largest PD GWAS (Chang et al. 2017) compared 6,476 PD cases with 302,042 controls, followed by a meta-analysis with over 13,000 PD cases and 95,000 controls. The authors however used an older GWAS dataset with lower case numbers (Nalls et al., 2014).

We thank the reviewer for pointing this out. A majority of our analysis was performed before Chang et al. 2017 was published. Unfortunately, due to time in journal reviewing process this has taken a while for our initial review. We therefore use Chang et al. 2017 to validate our TWAS results. We found that 21 of 66 unique TWAS genes were in significant GWAS loci from Chang et al. 2017, which nicely confirms our TWAS results (Supplementary Table S1). Another caveat is that the complete summary statistics for Chan et al. 2017 is not widely available due to a restriction on the 23andMe dataset. We have requested this data but still don't have access to the summary statistics. Waiting for data will significantly delay our publication.

The contribution of our manuscript is not only the prioritizing novel genes but also identify potential mechanisms (e.g., splicing and expression effect) and pathways (lysosomes in myeloid cells). Thus, we think the results are valuable for further experimental validation work. For these reasons, we would like to disseminate the results to the community as soon as possible.

(6) The Z-score for SNCA in PD association with RNA expression in brains and Z-score for SNCA PS association with peripheral monocyte RNA expression point in opposite directions, suggesting reduced alpha Synuclein expression in brains of PD patients. This is completely at odds with our current concept of PD pathophysiology. Please explain.

We thank the reviewer for pointing this out. The SNCA is one of the genes that were identified with both total expression and splicing. It turns out that in our case, the SNCA signal that we highlighted is the splicing effect (intronic splicing event: chr4:90757680:90757894) not the total expression effect. We reported the Z-score from the splicing association rather than total gene expression since the splicing effect explained the SNCA association. Thus, there is no contradiction between our results and the expectation that SNCA expression level Z-scores is in the same directions in monocyte and brain.

(7) The authors use the SNP rs17665188 for the H1/H2 polymorphism of MAPT. Nalls et al., however reported rs17649553 as primary PD-associated MAPT variant. Why did you emphasize a different SNP?

We report the most significant SNP that is in LD with Nalls et al. We used the H1/H2 tag SNP rs17665188 that is strong LD ($R^2 = 0.97$; $D'1 = 1$) with Nalls et al. We have also cited a reference suggesting that rs17665188 tags H1/H2 Zabetian et al. 2007(pg 9.).

(8) The authors conclude that genes in suggestive PD loci (e.g., CTSB) were identified in their study, but not in PD GWAS before. This is not correct, since CTSB had already been reported in the Chang 2017 PD GWAS.

We thank the reviewer for point this out. We performed majority of our analysis before Chang et al. 2017 was published. In fact, we discovered CTSB by using the Nalls et al. 2014 GWAS summary stat, where the SNPs had suggestive association signal (Supplementary Figure S43). Therefore, we think that our study did identify CTSB as a candidate PD gene. We think that Chang et al. 2017 nicely validates some of our findings from Nalls et al. 2014. We used the Chang et al. 2017 data for validation of our TWAS results. Nevertheless, we have revised our manuscript to highlight that CTSB was recently identified by Chang et al. 2017.

Reviewer #1 (Remarks to the Author):

The authors have performed a tremendous amount of work and I truly commend them for their efforts. The replication of expression and splicing prediction across independent datasets (here ROSMAP & CMC) is particularly compelling. The revised manuscript is considerably strengthened.

The authors have addressed all previous major comments.

Minor Comments

1. The section describing the correlation between estimates of heritability between ROSMAP and CMC reports a “Pearson $\rho = 0.54$ ” and “Pearson $P = 0.43$ ”. I’m guessing this is supposed to be Pearson’s rho or r?

Reviewer #3 (Remarks to the Author):

The manuscript is now laid out in a clear and understandable manner. The supplementary material is more informative than in the first version.

I still see the following issues:

Reviewer 3, point 4:

The authors mentioned that three genes of the TWAS results i.e. LRRK2, GPNMB and GBP7 were previously shown to be differentially expressed in PD brains vs controls. The authors still not properly state this information in the manuscript text. Please provided this information and corresponding references.

Reviewer 3, point 5 and point 8:

It is worrisome that GWAS raw data from Chang, Nature Genetics, 2017 are not accessible to the authors. Not even the summary statistics without patient’s IDs are accessible. This GWAS was

published in a Nature Family Journal. Denial of access to data for skilled scientists prevents scientific progress. Maybe the editors can help to provide access to the Chang data. Ideally, your analysis should be performed on the most recent and largest GWAS dataset, not on the second-best.

The Chang GWAS has been published in 2017. The current manuscript will probably be published in 2019. Please include at least the published data from the Chang GWAS to the main text of your manuscript, as far as they are relevant for your findings. The currently available Chang GWAS should not only confirm your data but should be included in your main analysis. E.g., the association of CTSB and PD had already been identified by Chang in 2017. Please state in your main text that you validated the results on CTSB but that it was previously identified by Chang and thus is no new finding in PD genetics.

Title: Prioritizing Parkinson's Disease genes using population-scale transcriptomic data

Manuscript reference number: Nature Communications, Li et al. 2018

RESPONSE TO REVIEWERS' COMMENTS

We are grateful to the Editor for inviting us to submit a revised version of the manuscript. We thank the two reviewers for critical feedback, thoughtful recommendations, and support of this work. We have made the minor changes requested by the reviewers. All changes are highlighted in the manuscript text (in WORD .DOC) file.

REVIEWERS' COMMENTS:

Reviewer #1 (Remarks to the Author):

The authors have performed a tremendous amount of work and I truly commend them for their efforts. The replication of expression and splicing prediction across independent datasets (here ROSMAP & CMC) is particularly compelling. The revised manuscript is considerably strengthened.

The authors have addressed all previous major comments.

Minor Comments

1. The section describing the correlation between estimates of heritability between ROSMAP and CMC reports a "Pearson $\rho = 0.54$ " and "Pearson $P = 0.43$ ". I'm guessing this is supposed to be Pearson's rho or r?

Thanks for pointing this out. We corrected "P" to "r"

"The average *cis*-heritability estimates between the ROSMAP and CMC were strongly correlated across genes and intronic splicing (Pearson $r = 0.54$). The prediction accuracy between the two cohorts was also strongly correlated (Pearson $r = 0.43$ and 0.16 for R^2 and $R^2 h_g^2$, respectively) (Supplementary Figures 7-8)."

Reviewer #3 (Remarks to the Author):

The manuscript is now laid out in a clear and understandable manner. The supplementary material is more informative than in the first version.

We thank the reviewer for constructive feedback on our manuscript.

I still see the following issues:

Reviewer 3, point 4:

The authors mentioned that three genes of the TWAS results i.e. LRRK2, GPNMB and GBP7 were previously shown to be differentially expressed in PD brains vs controls. The authors still not properly state this information in the manuscript text. Please provided this information and corresponding references.

We added a sentence in the main text to properly state the replication information:

"Recent studies have shown that both *SNCA* and *LRRK2* are highly expressed in human microglia³⁹, and that the expression levels of these two genes are elevated in peripheral monocytes of PD patients compared to age-matched controls^{40,41}."

In addition, both *GPNMB* and *GBP7* were previously shown to be differentially expressed in PD brains vs controls⁴²."

Reviewer 3, point 5 and point 8:

It is worrisome that GWAS raw data from Chang, Nature Genetics, 2017 are not accessible to the authors. Not even the summary statistics without patient's IDs are accessible. This GWAS was published in a Nature Family Journal. Denial of access to data for skilled scientists prevents scientific progress. Maybe the editors can help to provide access to the Chang data. Ideally, your analysis should be performed on the most recent and largest GWAS dataset, not on the second-best.

We agree with the reviewer. We have tried to get the data for several months but given the data is generated by a 23andMe it takes Data Transfer Agreement to get access.

The Chang GWAS has been published in 2017. The current manuscript will probably be published in 2019. Please include at least the published data from the Chang GWAS to the main text of your manuscript, as far as they are relevant for your findings. The currently available Chang GWAS should not only confirm your data but should be included in your main analysis. E.g., the association of *CTSB* and PD had already been identified by Chang in 2017. Please state in your main text that you validated the results on *CTSB* but that it was previously identified by Chang and thus is no new finding in PD genetics.

We have clarified this in the main text. The *CTSB* was previously found to be genome-wide significant in PD GWAS meta-analysis³." (with citation to Chang et al.)

"Most of these were located in loci with suggestive associations in PD GWAS ($5 \times 10^{-8} < p < 1 \times 10^{-6}$, Supplementary Figures 11–49), and four genes (*CTSB*, *PDLIM2*, *GALC*, and *C8orf5*) were previously found to be genome-wide significant in PD GWAS meta-analysis³."